



# Characteristics of Aeolian sediments transported above a gobi surface

Zhengcai ZHANG, Yan ZHANG*, Kaijia PAN

[1] Key Laboratory of Desert and Desertification, Northwest Institute of Eco-environment and Resources, Chinese Academy of Sciences, Lanzhou 730000, China

*Correspondence to*: Yan Zhang (zhangyan@nieer.ac.cn)

**Abstract.** Most previous studies of aeolian sediment transport have focused on shifting sand surfaces. As a result, sediment transport above gobi (gravel) surfaces is still poorly understood. In this field study, we quantified this transport to provide important support for parameterizing aeolian sediment transport models. We found that the relationship between the Sorensen horizontal sediment transport ($Q_S$) and shear velocity ($u_*$) could be expressed as $Q_S = \rho_a u_*^3/g(1-u_{*t}^2/u_*^2)(\alpha+\gamma u_{*t}/u_*+\beta u_{*t}^2/u_*^2)$,

where $\alpha = -127.4$, $\beta = 714.4$, and $\gamma = 737.0$. The relationship between the vertical sediment transport ($F$) and shear velocity could be expressed as $F_d = C_K \rho_a(u_*^2-u_{*t}^2)$, where $C_K = 0.75$. Although $Q$ and $F$ on gobi surfaces can be expressed similarly to previous results (i.e., similar equation forms), the coefficients were much larger than those for a shifting sand surface; that is, sediment transport was higher above the gobi. This difference resulted from the larger sand transport rate and saltation height above a gobi surface, and the larger transport and higher saltation height were related to gravel cover and soil crusts on the

gobi surface.

## 1 Introduction

Sand transport processes control sand and dust emission from surfaces; for example, the impact of saltating particles on the surface frees particles from the surface, leading to the emission of more sand and dust. Sand transport is also affected by landscape types (e.g., desert, agriculture, desertified land, grassland) because of the different surface characteristics, and by

the near-surface wind velocity and direction. The transported sand and dust have severe effects on humans and the natural environment. For example, transported sand can bury railways and dust in the air can be inhaled, damaging human health, and can directly affect radiative forcing by scattering and absorbing solar radiation (Kinne and Pueschel, 2001), thereby affecting climate change.

Sand transport has been widely studied above shifting sand surfaces using a variety of mathematical functions (Table 1).

This transport can be divided into natural and anthropogenic. Natural sand transport results mainly from wind erosion of surfaces such as gravel deserts, sandy deserts, and agricultural land, and has been widely studied, producing many useful conclusions (Zhang et al. 2003). In contrast, anthropogenic sand transport results from human activities, such as the desertification caused by unsustainable agriculture (Zhang et al. 2003; Wang et al., 2021). It also results from the construction of unpaved roads (Etyemezian et al., 2003), disruption of gravel surfaces, city construction, and transportation (Chen et al.,

2019). Anthropogenically modified surfaces are important dust sources (Ginoux et al., 2010). Previous research indicated that





anthropogenic dust accounted for about 19% of the global dust emission (Chen et al., 2019). In contrast, there has been little research on gobi (gravel) surfaces. Ho et al. (2011) found that sand transport above a hard surface (similar to a gobi) differed from transport above a shifting sand surface; saltating particles rebounded more strongly and reached greater heights. However, detailed field measurements of sand transport above gobi surfaces has been rare (e.g., Zhang et al., 2021b).

**Table 1** List of the most commonly used sediment transport functions. Source: Kok et al. (2012). $M_z$ represents the mean particle diameter, $D$ represents the reference diameter, $\rho_a$ is the density of air, $u_*$ represents the shear velocity, $u_{*t}$ represents the threshold shear velocity, and $g$ represents the acceleration due to gravity.

| Study | Equation | Comments |
|---|---|---|
| Bagnold (1941) | $Q_B = C_B (M_z/D)^{1/2}\rho_a u_*^3/g$ | Where $C_B$ represents a scale coefficient, $C_B = 1.5, 1.8,$ and $2.8$ for uniform, naturally graded, and poorly sorted sand, respectively. $C_B = 47.5$ in the present study. |
| Kawamura (1951) | $Q_K = C_k\rho_a u_*^3 / g(1-u_{*t}^2/u_*^2)(1 + u_{*t}/u_*)$ | $C_K = 2.78$ (Kawamura 1951) or $2.61$ (White 1979). $C_K = 26.8$ in the present study. |
| Owen (1964) | $Q_O = \rho_a u_*^3 /g (1-u_{*t}^2/u_*^2)(0.25 + v_t/3u_*)$ | $v_t$ is a particle's terminal fall speed |
| Lettau & Lettau (1978) | $Q_L = C_L(D_p/D_{250})^{1/2}\rho_a u_*^3(1-(u_{*t}/u_*))/g$ | $C_L = 6.7$. $C_L = 82.4$ in the present study |
| Ungar and Haff (1987) | $Q_{UH} = C_{UH}\rho_a(D_p/g)^{1/2} u_*^2 (1-[u_{*t}^2/u_*^2])$ | Ungar and Haff (1987) did not estimate a value of $C_{UH}$. $C_{UH} = 28.4$ in the present study. |
| Sorensen (2004) | $Q_S = \rho_a u_*^3 / g(1-u_{*t}^2/u_*^2)(\alpha+\gamma u_{*t}/u_*+\beta u_{*t}^2/u_*^2)$ | Where $\alpha$, $\beta$, and $\gamma$ are parameters that characterize the dimensions of a typical saltation hop. $\alpha=-127.4$, $\beta=714.4$, and $\gamma=737.0$ in the present study. |
| Pähtz et al. (2011) | $Q_P = \rho_a u_*^2 / g(1-u_{*t}^2/u_*^2)(au_{*t}-b)$ | For uniform 250-μm sand, $a = 19$ and $b = 1.6$. $a = 81.5$ and $b = 3.5$ in the present study. |
| Kamath et al. (2022) | $Q_{KS} = b[1+c(u_*/u_{*t}-1)](M_z /g)^{1/2} \rho_a(u_*^2-u_{*t}^2)$ | $b$ and $c$ depend on the sand cover thickness. $b= 81.5$ and $c = 0.12$ in the present study. |

Similarly, vertical sediment transport has been widely studied (Table 2). Dust transport is controlled by factors such as the gravel cover, vegetation cover, silt and clay contents, and presence or absence of a soil crust (Zhang et al., 2017b; Cui et al.,

2019). Gravel, vegetation, and a soil crust can protect the surface by increasing its cohesion and acting as a roughness element, which means that dust is more difficult to entrain because of the increased threshold shear velocity (Zhang et al., 2021a). However, when these surface elements are significantly disturbed, dust emission can occur through exposure of the underlying silt and clay under strong wind erosion (Belnap and Warren, 2002; Goossens and Buck, 2009; Bullard et al., 2011), and these changes decrease the threshold shear velocity (Zhang et al., 2008). Surface disturbances are mainly caused by animal trampling,





cultivating the soil for agriculture, and vehicle traffic, all of which can greatly increase dust emission. Tegen et al. (2004) indicated that human activities reduced the threshold velocity for erosion of cultivated soils, with human disturbances making the soil more susceptible to erosion. Baddock et al. (2011) indicated that trampling of clay-rich dry lake soil crusts by animals obviously increased dust emission. For a gravel surface, Meng et al. (2019) indicated that surface compaction by vehicles can increase dust emission to between 5 and 50 times the emission from an undisturbed surface. All these studies emphasized that

human activities greatly effect sand transport processes, but because few researchers have studied gobi surfaces, the effects of human activities on sand transport above gobi surfaces is unclear.

**Table 2** List of the most commonly used vertical sediment transport functions. $M_z$ represents the mean particle diameter, $D$ represents the reference diameter, $\rho_a$ is the density of air, $u_*$ represents the shear velocity, $u_{*t}$ represents the threshold shear velocity, and $g$ represents the acceleration due to gravity.

| Study | Equation | Comments |
|---|---|---|
| Kawamura (1951) | $F_d = C_K \rho_a u_*^3 / g(1 - u_{*t}^2/u_*^2)(1 + u_{*t}/u_*)$ | $C_{Ka}$ has units of m$^{-1}$. $C_{Ka}$=6.87 in the present study. |
| Ungar and Haff (1987) | $F_{UH} = C_{UH} \rho_a (D_p/g)^{1/2} u_*^2 [1 - (u_{*t}^2/u_*^2)]$ | $C_{UH}$=7.25 in the present study. |
| Gillette and Passi (1988) | $F_d = C_{GP} u_*^4 (1 - u_{*t}/u_*)$ | $C_{GP}$ has units of kg m$^{-6}$ s$^3$. $C_{GP}$=2.08 in the present study. |
| Shao et al. (1993) | $F_d = C_S \rho_a u_* (u_*^2 - u_{*t}^2)$ | $C_S$ has units of s$^2$/m$^2$. $C_S$=0.98 in the present study. |
| Marticorena and Bergametti (1995) | $F_d = aQ$ | $a$ is the *sandblasting efficiency*, which ranges between 10$^{-5}$ and 10$^{-2}$ m$^{-1}$. $a$ =0.25, $R^2$=0.84, $RMSE$=0.09 in the present study. |
| Kok et al. (2012) | $F_d = C_K \rho_a (u_*^2 - u_{*t}^2)$ | $C_K$ has units of s$^3$/m$^3$. $C_K$=0.75 in the present study. |

In northwestern China, gravel deserts (gobis) have been described as a "wide, shallow basin of which the smooth rocky bottom is filled with sand, silt or clay, pebbles or, more often, with gravel" (Cooke, 1970). These surfaces cover 56.9×10$^4$ km$^2$ in northern China and are a major landscape feature. Although the gravel cover is more than 50% for most gravel desert surface, and soil crusts cover more than 50% of the total land surface (Zhang et al., 2021a), road construction, city construction, and human activities disturb the gravel surface and crusts, and have disturbed most gravel surfaces to at least some extent.

Wind tunnel studies indicated that the annual average contributions of China's gobi deserts to PM$_{10}$ emission totaled 6.1 Tg yr$^{-1}$ from the 1970s to 2015, and this made gobis the main dust source areas in northern China (Wang et al., 2021). Zhang et al. (2003) found that dust emission from northern China amounted to 22% of the total global dust emission. In 2021, dust storms in gobi regions increased obviously, and have received much research attention (Filonchyk, 2022). However, field measurement of sand and dust transport above gobi surfaces is scarce (Zhang et al., 2021a), and its mechanisms are still not

well understood. To provide some of the missing knowledge, we designed the present study to quantify sand transport rates and the grain-size distribution on gobi surfaces using field measurements, and describe the mechanism of sand transport above



gobi surfaces. We hypothesized that the characteristics of sand transport and the underlying mechanisms for gobi surfaces would differ from those for sandy surfaces.

## 2 Methods and material

### 2.1 Study region

China's Alxa Plateau is the gobi region with the strongest dust storms (Han et al., 2012; Xu et al., 2020). This is partially due to the low rainfall (annual precipitation is typically less than 100 mm), combined with low vegetation cover (typically <5%). The annual average temperature is 8.3°C, with mean monthly temperatures ranging from –9.7°C in January to 23.1°C in August. Annual average wind speed is 4.4 m s$^{-1}$. However, winds are strong enough to cause dust storms (>17 m s$^{-1}$) on 40 to 50 days per year, with most dust storms occurring during the winter and spring, when precipitation and vegetation cover are both low.

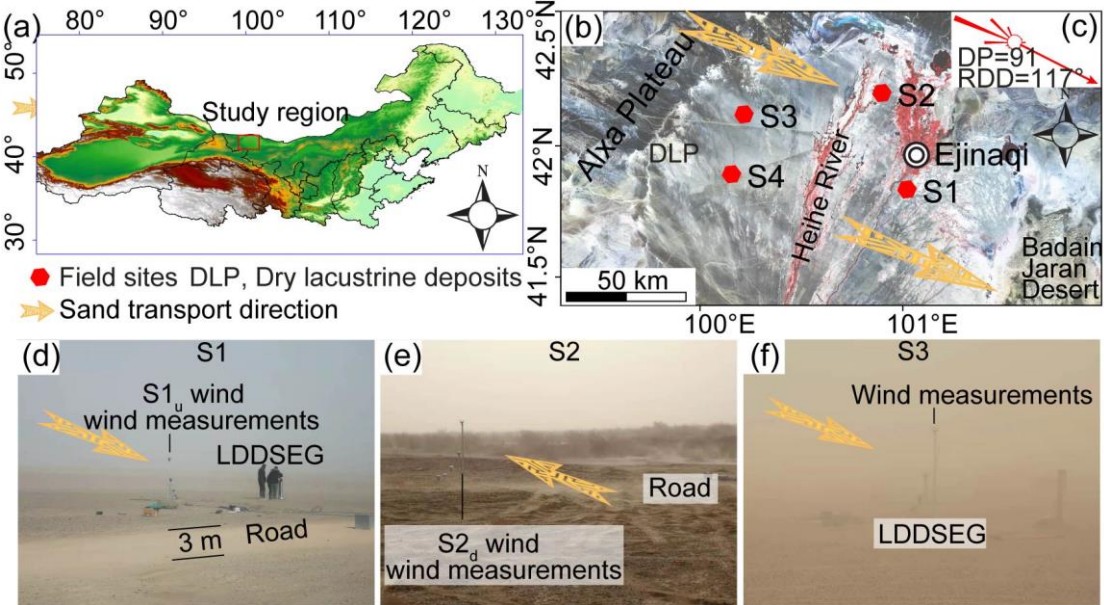

**Figure 1** (a) Location of the study region, (b) location of field experiment sites, (c) the potential sand transport (*DP*, drift potential; *RDD*, resultant drift direction), and images of the study plots at (d) the S1 field site, (e) the S2 field site, and (f) the S3 field site. (g) the layout at the at S4 field site. "u" and "d" represent upwind and downwind sites, respectively. Field measurement layouts showed in Appendix A1.

### 2.2 Methods

The field data were collected from 10 to 14 January 2021 at four sites in the area where strong dust storms occurred. During this period, almost all of northwestern China was affected by a dust storm, so our results are representative of dust transport in this region.



### 2.2.1 Dust transport

To collect details of the sediment transport characteristics over the gobi surface, we used the LDDSEG vertical segmented sediment sampler to continuously measure sediment transport to a height of 1 m. The sampler was designed by the Key Laboratory of Desert and Desertification, Chinese Academy of Sciences, and is widely used in sediment transport measurements in China. Based on wind tunnel tests, the sampler captures 86% of the particles being transported below a height of 1 m above the surface. For details about this sampler and the field measurement method, see Zhang et al. (2021b). The sampler collected blowing sand in 0.02 m × 0.02 m sections. The collected sediment was weighed using an electronic balance with a precision of 1 mg. Field measurements were obtained during seven periods during the severe dust storm that occurred from 10 to 14 January 2021, and were labeled $S1_u$, $S1_d$, $S2_u$, $S2_d$, $S3_1$, $S3_2$, $S4_u$, and $S4_d$, where "u" and "d" refer to upwind and downwind samples, respectively (Fig. 1d-g).

### 2.2.2 Grain size measurements

We determined the particle-size distribution of the collected sediments using a Malvern MasterSizer 3000 (Malvern Instruments Ltd., Malvern, England) at the Key Laboratory of Desert and Desertification, Chinese Academy of Sciences, in Lanzhou. One surface sample was collected at each site and all samples of the wind-transported sand (at 50 heights) were analyzed. Samples were split using a microsample splitter to minimize the bias. We divided the grain size into six categories: $PM_{10}$ (<10 μm), clay and silt (< 63 μm), very fine sand (63 to 125 μm), fine sand (125 to 250 μm), medium sand (250 to 500 μm), and coarse sand (>500 μm).

### 2.2.3 Climate conditions

Climate data was obtained from automatic weather stations at Ejinaqi (Fig. 1b) to describe the regional climate. All data were obtained from China's National Climatic Data Center (https://data.cma.cn/en). All sensors were set at 10 m above the ground in accordance with World Meteorological Organization standards for anemometer heights. Appendix A2 presents the wind speed, wind direction, air temperature, air relative humidity, and air pressure data from 10 January 2021 to 14 January 2021 during the 4-day dust storm event.

Wind velocity and wind direction were measured on four different days for the four sites during the field measurements (Fig. 2). Air temperature and relative humidity were measured using a CS215 meter (Campbell Scientific Inc., Logan, UT, USA). Wind velocity and directions were measured using Windsonic sensors (Gill Instruments Limited, Lymington, UK). Data were recorded at 1-s intervals and stored as 1-min averages. Data were stored in CR6 dataloggers (Campbell Scientific). Mean wind velocity was lowest (8.1 ± 2.2 m s$^{-1}$) at site S2 on 11 January 2021 and highest (13.2 ± 2.6 m s$^{-1}$) at site S3 on 12 January 2021. The wind direction was from the northwest to the southeast (a mean azimuth of 291 ± 13º to 347 ± 8º at the four sites) (Fig. 2).



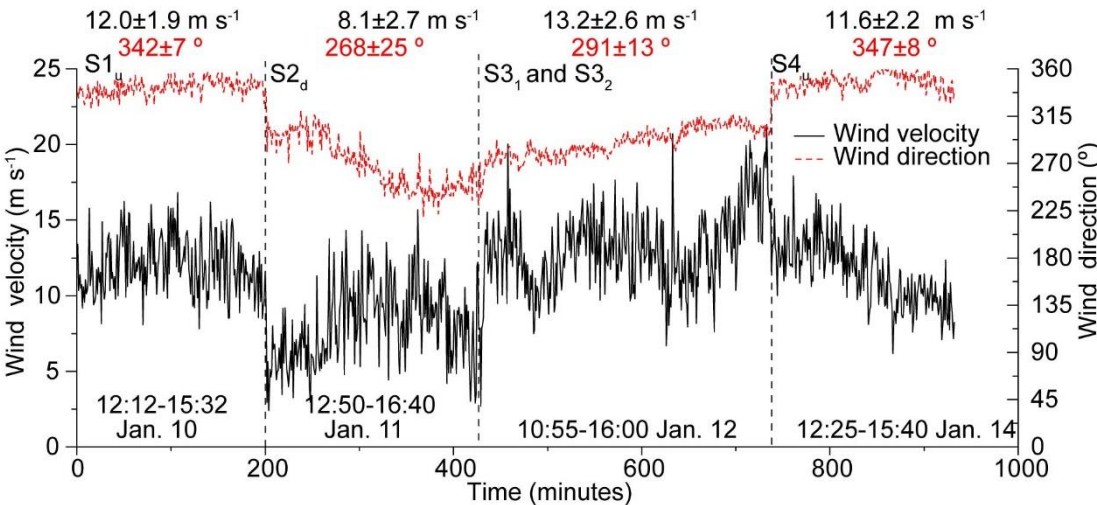


**Figure 2** Wind velocity and direction (azimuth) during the field experiments. Site locations are shown in Figure 1.

### 2.3 Data analysis

We calculated the threshold wind velocity ($u_{*t}$) using the method of Shao et al. (1996):

$$u_{*t} = u_{*t0}RHM \qquad (1)$$

where $R$, $H$, and $M$ are functions that describe the influences of surface roughness, soil moisture, and a soil crust, respectively, and threshold shear velocity ($u_{*t0}$) is calculated as follows:

$$u_{*t0} = a_1(\sigma_p gd + a_2/\rho d)^{0.5} \qquad (2)$$

where $d$ is the mean diameter of the erodible grains; $g$ is the acceleration due to gravity; $\rho$ is the air density; and $\sigma_p$ is the particle-to-air density ratio. For more details, see Zhang et al. (2021a). Coefficients $a_1 = 0.0123$ and $a_2 = 3 \times 10^{-4}$ kg s$^{-2}$ were

obtained from Shao and Lu, (2000). Table 4 shows the calculated threshold shear velocities.

**Table 4** The calculated threshold shear velocities: fluid threshold shear velocity ($u_{*t0}$) and threshold wind velocity ($u_{*t}$).

|  | S1$_u$ | S1$_d$ | S2$_u$ | S2$_d$ | S3$_1$/S3$_2$ | S4$_u$ | S4$_d$ |
|---|---|---|---|---|---|---|---|
| $u_{*t0}$ (m s$^{-1}$) | 0.17 | 0.17 | 0.19 | 0.16 | 0.20 | 0.21 | 0.18 |
| $u_{*t}$ (m s$^{-1}$) | 0.32 | 0.34 | 0.35 | 0.30 | 0.37 | 0.38 | 0.34 |

We calculated the time-averaged horizontal transport rate at height $z$ above the surface ($q_z$, kg m$^{-1}$ h$^{-1}$) by dividing the mass of collected sediment at each height by the inlet area (2 cm $\times$ 2 cm = 4 cm$^2$) perpendicular to the wind direction. We then fit these data to a Gaussian peak function, which was used to express sediment transport over a gobi surface (Zhang et al., 2017a,

2021b):

$$q_z = a_1 + q_0 \exp(-0.5[|z - z_q|/a_2]^2) \qquad (3)$$

where $a_1$, $q_0$, $z_q$, and $a_2$ are the regression coefficients. $q_0$ is a scaling parameter for the profile, and $z_q$ is the height where the maximum sediment transport rate occurs.




The total transport ($Q_T$, kg m$^{-1}$ h$^{-1}$) was calculated as follows:

$$Q_T = \sum_{i=1}^{50} q_z \qquad (4)$$

Where $q_z$ represents the sediment transport at height $z$ in sediment collection chamber $i$.

The vertical profile for $u_z$, the horizontal wind velocity (m s$^{-1}$) at height $z$ (m), can be described by applying the law of the wall (Bagnold, 1941):

$$u_z/u_* = \ln(z/z_0) \, / \, k \qquad (5)$$

where $u_*$ is the shear velocity (m s$^{-1}$), $z_0$ is the aerodynamic roughness length (m), and $k$ is Von Karman's constant (0.4). The relationship between wind velocity and height can be expressed as a log-linear function (Dong et al., 2003):

$$u_z = a + b \, \ln z \qquad (6)$$

where $a$ and $b$ are regression coefficients.

$u_*$ and $z_0$ can be calculated by the gradient method or the wind profile method, which produce similar results (Zhang et al., 2004). We chose the wind profile method:

$$z_0 = \exp(-a \, / \, b) \qquad (7)$$

$$u_* = k \, b \qquad (8)$$

We calculated the vertical dust emission ($F$, kg m$^{-1}$ h$^{-1}$) using the method of Gillette et al. (1972) for the collected sand samples:

$$F = k u_* \bar{z} \frac{c_2 - c_1}{z_2 - z_1} \qquad (9)$$

where the subscripts 1 and 2 refer to the horizontal aeolian flux in traps 1 and 2 at each of the four sites (0.99 and 0.07 m), respectively. $c$ is the horizontal concentration of transported sediment (kg m$^{-1}$ h$^{-1}$), $k$ is the von Karman constant (0.40), $u_*$ is the friction velocity (m s$^{-1}$), and $\bar{z}$ is the mean height (the mean of $z_1$ and $z_2$, which equaled 0.46 m for S1, 0.49 m for S2, and 0.48 m for the others). $F_s$ is the emission rate for all sediment, and $F_{10}$ is the PM$_{10}$ emission rate.

To evaluate the goodness of fit of these equations with the empirical data, we used the root-mean-square error (*RMSE*).

## 3 Results

### 3.1 Wind velocity during the four field measurement periods

Land surface properties (e.g., gravel cover, silt and clay contents, soil physical crust) can affect the near-surface wind velocity, and can therefore affect dust emission. We successfully expressed the wind profiles as log-linear functions (Fig. 3). The calculated $z_0$ ranged from $0.76 \times 10^{-3}$ to $0.81 \times 10^{-3}$ m for undisturbed gobi surfaces, and from $0.70 \times 10^{-3}$ to $0.76 \times 10^{-3}$ m on disturbed gobi surface for wind velocity ranging from 12 to 13 m s$^{-1}$ (Table 5). This indicated that wind erosion occurred more easily on disturbed gobi surfaces than undisturbed gobi surfaces. However, $u_*$ was similar at site 1 (S1) for the disturbed and undisturbed gobi surfaces.

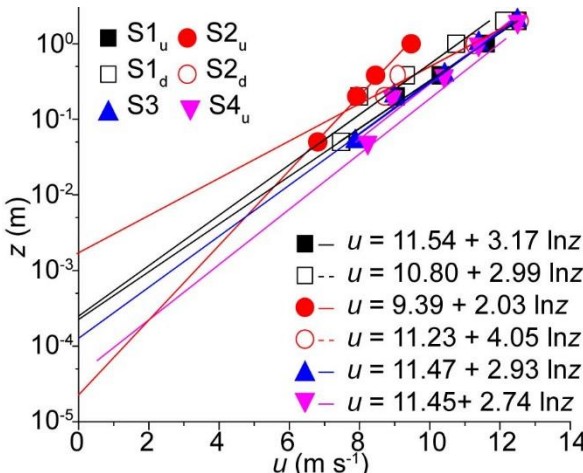

**Figure 3** Mean wind profiles ($u$ ranged from 12 to 13 m s$^{-1}$) for the field experiment sites (Fig. 1d-g). $R^2 > 0.96$, $P < 0.001$ for all regressions. S3 represents the mean values for subsites S3$_1$ and S3$_2$. No data was available for S4$_d$.

**Table 5** The aerodynamic roughness length ($z_0$, m), calculated shear velocity ($u_*$, m s$^{-1}$), maximum sediment transport height ($T_h$, m), total transport rate ($Q_T$), and mean transport rate ($Q_m$, is the mean $Q_T$ during each measurement period) at the four field measurement sites.

| | | $z_0$ ($\times 10^3$ m) | $u_*$ (m s$^{-1}$) | $T_h$ (m) | $Q_T$ (kg m$^{-1}$ h$^{-1}$) | $Q_m$ (kg m$^{-1}$ h$^{-1}$) |
|---|---|---|---|---|---|---|
| S1$_u$ | Undisturbed | 0.76 | 1.23 | 0.07 | 30.12 | 0.60±0.61 |
| S1$_d$ | Disturbed | 0.76 | 1.20 | 0.03 | 60.73 | 1.21±1.43 |
| S2$_u$ | Undisturbed | 0.81 | 0.81 | 0.00 | 8.21 | 0.16±0.03 |
| S2$_d$ | Disturbed | 0.70 | 1.62 | 0.05 | 15.82 | 0.32±0.21 |
| S3$_1$ | Undisturbed | 0.77 | 1.17 | 0.05 | 70.98 | 1.42±0.82 |
| S3$_2$ | Undisturbed | 0.77 | 1.17 | 0.05 | 149.43 | 2.99±2.34 |
| S4$_u$ | Undisturbed | 0.79 | 1.10 | 0.07 | 20.24 | 0.40±0.43 |
| S4$_d$ | Disturbed | No data | No data | 0.07 | 13.54 | 0.27±0.31 |

**3.2 Horizontal sediment transport flux**

Sand transport quantities differed among the field measurement sites (Fig. 4a). At S1, $Q_T = 30.12$ kg m$^{-1}$ h$^{-1}$ at the upwind end and 60.73 kg m$^{-1}$ h$^{-1}$ at the downwind end (Table 5), which is about 2.02 times the upwind value. However, at S4, $Q_T$ at the upwind end was about 1.49 times that at the downwind end. Below a height of 0.1 m, some transport curves (except S2$_u$) showed a slight increase in transport before following the expected decrease with increasing height (Fig. 4a). Except for the upwind sample at S2 (a dry lacustrine deposit surface), all sand transport quantity curves revealed a threshold height (0.03 m

to 0.15 m) above the gobi surface (Table 5). The threshold height ($T_h$) for maximum sand transport decreased from upwind to downwind at S1, increased at S2, and did not change at S3 and S4 (Table 5). $T_h$ is related to the gravel cover and wind velocity. For example, at S1, the mean wind velocity was similar between $S1_u$ and $S2_d$ (12.9 m s⁻¹), although $S1_u$ had a gravel surface, but $S1_d$ was rich in silt and clay but had almost no gravel; as a result, $T_h$ was larger at $S1_u$ than at $S1_d$ (Table 5). At S3, the

landscape was the same at both subsites, but the wind velocity was larger at $S3_d$ than at $S3_u$ and $T_h$ (0.05 m) did not differ between the subsites (Table 5). The sand transport flux can be expressed as a Gaussian peak function ($q_z = b_1 + q_0$ exp (-0.5[|z − $z_q$|/$b_2$]²)) (Table 6), which was similar to the results of Zhang et al. (2021b).

**Table 6** The coefficients of hozirontal sediment transport flux used in equation 3 for $S1_u$, $S1_d$, $S2_d$, $S3_1$, $S3_2$, $S4_u$, and $S4_d$. $S2_u$ followed an exponential function ($q_z = b_1 + q_0$ exp ($z/b_2$)²). $P<0.02$ for all regressions.

|        |             | $b_1$ | $q_0$ | $z_q$ | $b_2$ | $R^2$ | RMSE |
|--------|-------------|-------|-------|-------|-------|-------|------|
| $S1_u$ | Undisturbed | 0.29  | 1.79  | 0.08  | 0.08  | 0.91  | 0.18 |
| $S1_d$ | Disturbed   | 0.49  | 4.15  | 0.06  | 0.10  | 0.91  | 0.44 |
| $S2_u$ | Undisturbed | 0.40  | 65.14 |       | 0.16  | 0.96  | 0.01 |
| $S2_d$ | Disturbed   | 0.21  | 0.67  | 0.02  | 0.12  | 0.97  | 0.03 |
| $S3_1$ | Undisturbed | 0.94  | 2.26  | 0.06  | 0.12  | 0.93  | 0.21 |
| $S3_2$ | Undisturbed | 1.42  | 6.39  | 0.04  | 0.17  | 0.94  | 0.58 |
| $S4_u$ | Undisturbed | 0.14  | 1.19  | 0.06  | 0.13  | 0.96  | 0.09 |
| $S4_d$ | Disturbed   | 0.08  | 0.90  | 0.03  | 0.14  | 0.95  | 0.07 |

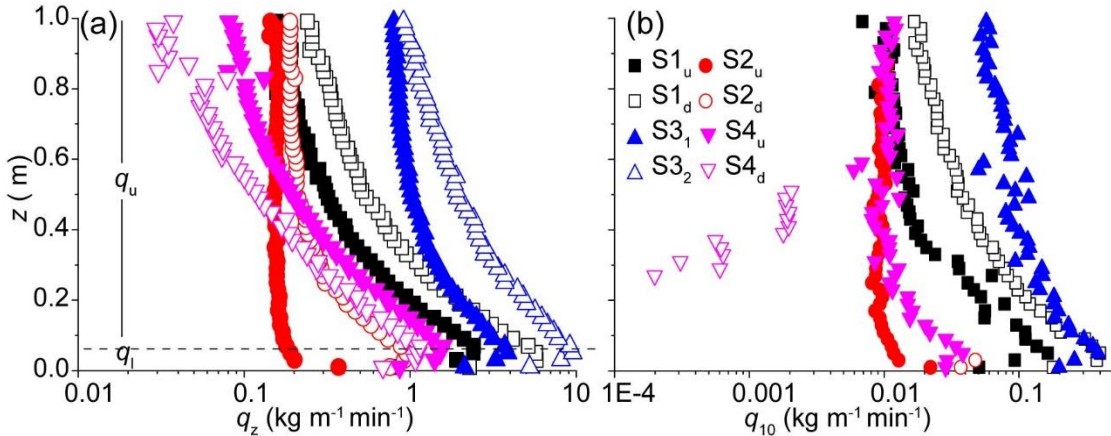


**Figure 4 (a)** Horizontal sediment transport rate ($q_z$) and (b) change in PM₁₀ transport ($q_{10}$) with height ($z$).

Figure 4b shows that the pattern for the dust transport ($q_{10}$) was similar to that for the sand transport (Fig. 4a), with transport increasing to a certain height before stabilizing or decreasing continuously. However, the change in the dust transport rate was much slower than for sand.


Figure 4a shows that the rate of change of sediment transport with height differed among the four sites. We used the ratios of the mean sediment transport above and below $T_h$ ($q_u/q_l$, respectively) to express these differences. The ratio decreased with height: for $S1_u$ and $S1_d$, the ratio was 3.0 and 5.9, respectively; for $S2_u$ and $S2_d$, the ratio was 21.2 and 4.7, respectively; for $S3_1$ and $S3_2$, the ratio was 6.7 and 5.5, respectively; and for $S4_u$ and $S4_d$, the ratio was 3.0 and 5.9, respectively.

### 3.3 Vertical dust flux above the gobi surface

The vertical sediment transport ($F_s$) showed an obvious difference between sites (Fig. 5a). $F_s$ was largest at $S3_2$ (0.70 kg m$^{-1}$ h$^{-1}$) and smallest at $S2_u$ (0.02 kg m$^{-1}$ h$^{-1}$). $F_s$ averaged 0.23±0.25 kg m$^{-1}$ h$^{-1}$ across all sites, and $S1_d$ and $S3_1$ were comparable to the values reported by Zhang et al. (2021b), with a mean of 0.41±0.20 kg m$^{-1}$ h$^{-1}$ (Fig. 5a). For S2, scattered dwarf shrubs with vegetation cover of 20% grew at the upwind edge of the field site, therefore, sand availability was limited, and this explains why Fs had its minimum value at this site. $F_s$ for the disturbed gobi surface ($S1_d$, $S2_d$, $S4_d$) was 1.6 to 2.1 times the value above the undisturbed gobi surface ($S1_u$, $S2_u$, $S4_u$), which indicated that disturbed gobis can provide more vertical sediment flux during dust storms.

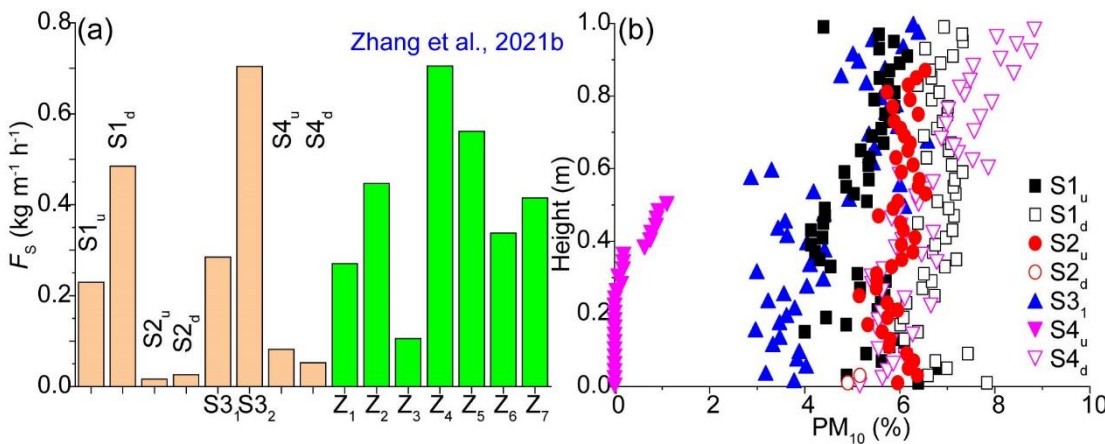

**Figure 5** (a) The vertical dust emission rates above the gobi surface. (b) The change of the PM$_{10}$ concentration with height on gobi surface.

### 3.4 The transported PM$_{10}$ concentration

The PM$_{10}$ transport rate also decreased with height (Fig. 5b), though following a different pattern from sand (Fig. 4a). The largest transport occurred at 0.05 m above the surface. PM$_{10}$ changed with height in complicated ways (Fig. 5b). The mean transported PM$_{10}$ was 3.9±2.4% for all field measurement sites. The transported PM$_{10}$ was larger above disturbed gobi surfaces than undisturbed gobi surfaces (Fig. 5b), and reached as high as 6.8±0.4% ($S1_d$), and as low as 0.3±0.4% ($S4_u$). If we compare

the PM$_{10}$ transported above shifting sand (Zhang et al., 2017b) with that transported above the gobi surfaces in the present study, dust transport above the gobi surface was 4 times the transport above shifting sand (Fig. 6a). PM$_{10}$ transport rates did not affect the PM$_{10}$ proportion (Fig. 6b).





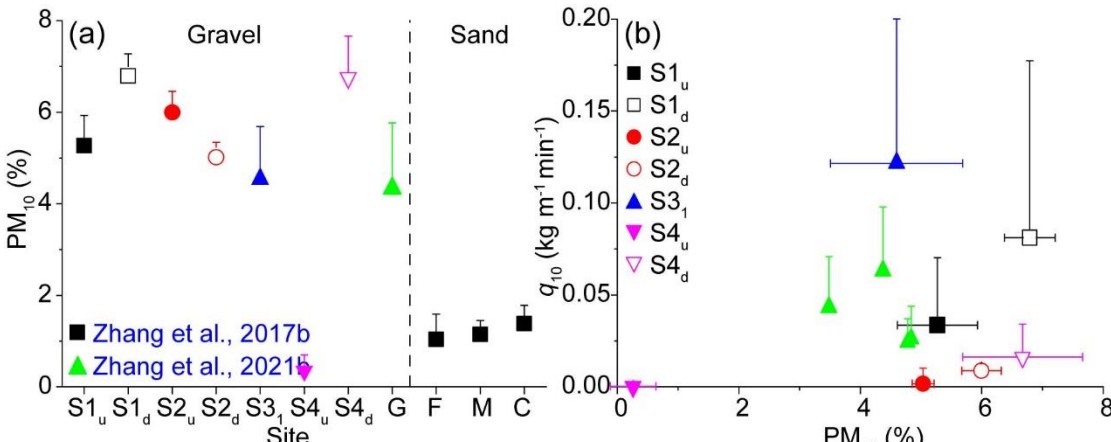

**Figure 6** (a) $PM_{10}$ concentration collected by the sand samplers. G, gobi desert data from Zhang et al. (2021b); F, fine sand,
M, medium sand, C, coarse sand data from Zhang et al. (2017b). (b) Relationships between the transport ($q_{10}$) and proportion
of $PM_{10}$.

### 3.5 Grain-size distribution of transported aeolian sediment

Figures 7, Appendix A3, and A4 show how the grain-size distribution changed with height at the four field experiment sites.
During these dust storms, the transported sediment was mainly sand >125 μm in diameter, for which values to a height of 1.0
m above the surface ranged from 40.2±13.3% (mean ± SD) to 70.8±5.6%, with a mean of 51.1±11.3%, followed by very fine
sand from 63 to 125 μm, with values to a height of 1.0 m above the surface ranging from 23.2±5.8% to 32.9±5.7% and with a
mean of 28.0±3.7%. The silt component had values ranging from 5.5±3.6% to 26.6±9.8%, with a mean of 19.6±8.5%, and the
clay components had values ranging from 0 to 2.0±0.2%, with a mean of 1.4±0.6%.

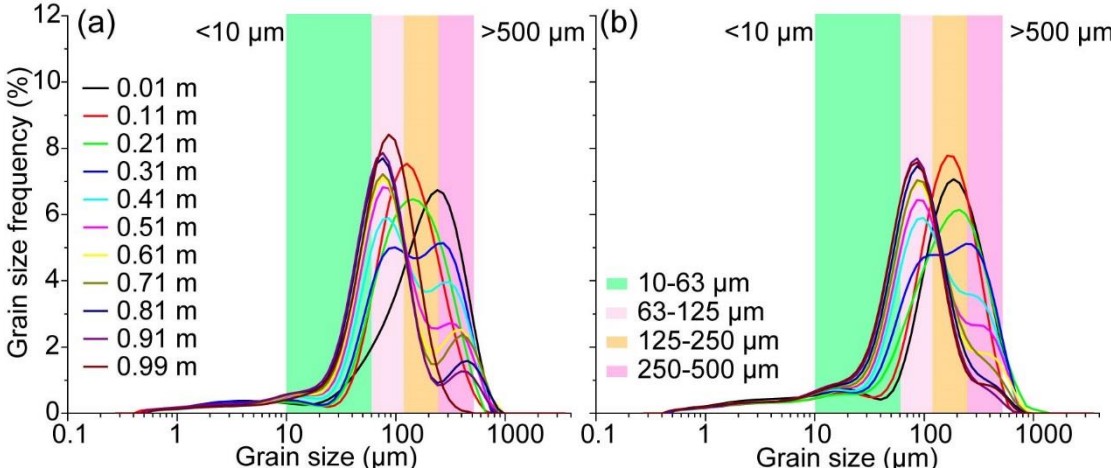

**Figure 7** The grain size distributions at site S1: (a) upwind, (b) downwind. Curves for the other sites are shown in Figure S2.





## 4. Discussion

### 4.1 Relationships between the mean sand transport rate above the gobi surfaces ($Q_m$) and wind velocity

Since Bagnold (1941), the relationships between the mean sand transport rate ($Q_m$) and wind velocity or wind shear velocity have been widely studied (Table 1), but these studies almost all focused on shifting sand surfaces (Kok et al., 2012). In

summary, sand transport rates were related to $u_*/u_{*t}$ (Kok et al., 2012). Based on our study and previous research (Zhang et al., 2021b), we found that $Q_m$ predicted the dimensionless sand transport rate ($\hat{Q}$) well (Fig. 8a):

$$\hat{Q} = Q_m / (\rho_p d \ ([\rho_p / \rho_a -1]gd)^{0.5} \tag{10}$$

where $\rho_p$ is 2650 kg/m$^3$, $g$ is 9.81 kg m$^{-3}$, and $\rho_a$ is 1.25 kg/m$^3$). In addition, the Shields number also predicted ($\hat{Q}$) well (Fig. 8b):

$$\Theta = \rho_a u_*^2/([\rho_p-\rho_a]gd) \tag{11}$$

$\hat{Q}$ was also predicted well by the dimensionless shear velocity ($u_*/u_{*t}$) (Fig. 9a). The calculated $\hat{Q}$ ranged from 0.5 to 13.8 (4.38±3.67) (Fig. 9a). These results further indicated that the sand transport rate above a gobi surface was much larger than that above a shifting sand surface.

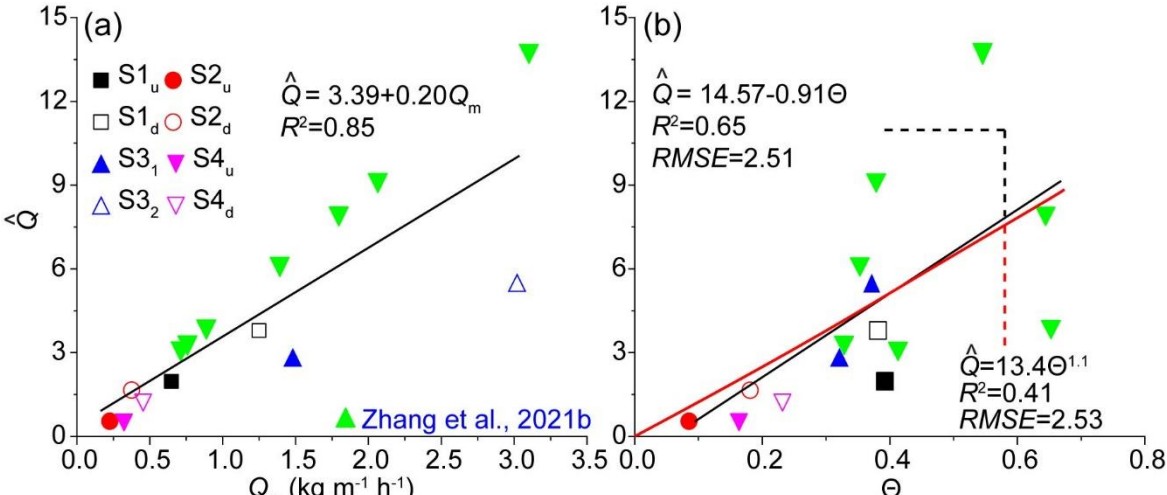

**Figure 8** (a) The relationship between dimensionless horizontal sediment transport ($\hat{Q}$) and measured sediment transport ($Q_m$). (b) The relationships between dimensionless horizontal sediment transport ($\hat{Q}$) and the Shields number ($\Theta$). $P<0.05$.





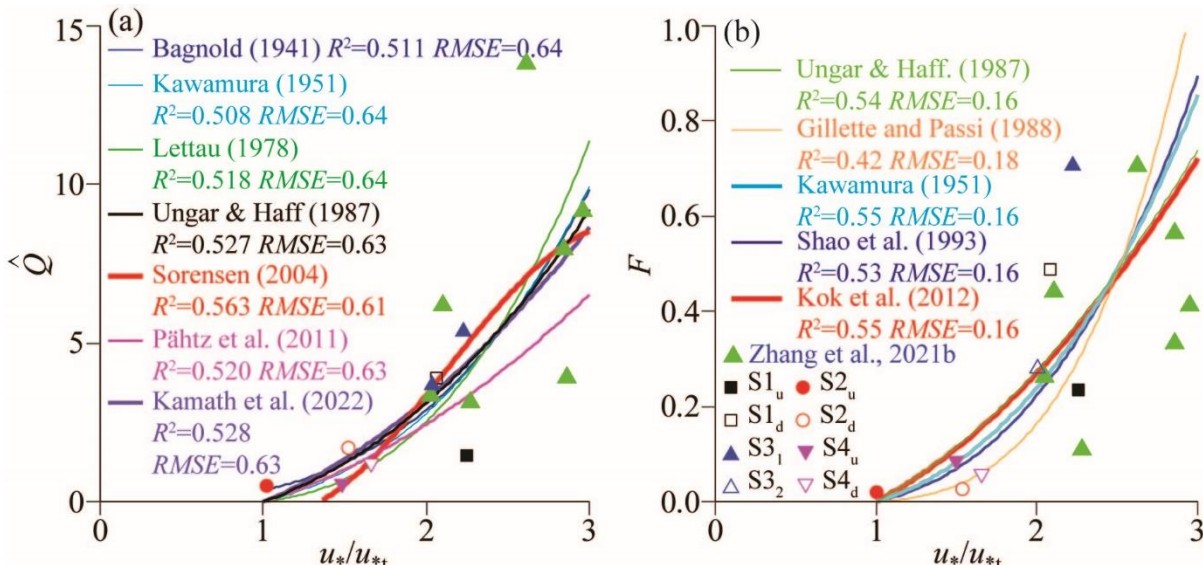

**Figure 9** (a) The dimensionless horizontal sediment transport ($\hat{Q}$) predicted by the functions listed in Table 1, where $u_*/u_{*\mathrm{t}}$ represents the dimensionless shear velocity. Table 1 shows the coefficients calculated in this study. (b) The vertical sediment

245            transport ($F$) predicted by the functions listed in Table 2. $P<0.05$ for all regressions.

Using the equations in Table 1, we found that Sorensen's (2004) function provided the best fit to our data, followed by the function defined by Kamath et al. (2022) (Fig. 8a). However, all regression coefficients between $\hat{Q}$ and $u_*/u_{*\mathrm{t}}$ were larger (Table 1) than those above a shifting sand surface (Kok et al., 2012). Previous research indicated that $\hat{Q}$ and $\Theta$ for a water-eroded gravel surface can be expressed as $\hat{Q} \propto \Theta^{2.5}$ (Paintal, 1971). However, our field data showed that $\hat{Q}$ increased with $\Theta$

as a linear or power function (Fig. 8a).

**4.2 Relationships between the vertical sand transport rate ($F$) and wind velocity above the gobi surface**

Different land surface types (sand dunes, playas, alluvial fans, sandy gravel, gobi deserts, and abandoned land) have different dust emission (Wu et al., 2018). For gobis, the undisturbed land surface covered by gravel and surfaces with a biological or chemical crust have less available dust (Zhang et al., 2016), but when the gobi surface is disrupted by humans, cars, or animals,

the underlying silt and clay are exposed to the wind, thereby increasing dust availability for transport (Fig. 1d, 4b). This led to emission 1.7 to 8.9 times that from an undisturbed surface. Surface disturbance not only increases the total sand transport (Fig. 4), but also the $PM_{10}$ concentration (Fig. 6).

Both the vertical sand transport rate ($F$) and the $PM_{10}$ transport rate ($F_{10}$) increased with increasing $u_*/u_{*\mathrm{t}}$ (Fig. 9b, 10a). $F$ and $u_*/u_{*\mathrm{t}}$ can be expressed using the functions of Kawamura (1951) and Kok et al. (2012) with a moderate goodness of fit

(Fig. 9b, $R^2 \geq 0.55$, $RMSE \leq 0.16$; Table 2). $F_{10}$ and $u_*/u_{*\mathrm{t}}$ can be expressed using the functions of Kawamura (1951) and Kok et al. (2012), but with a low goodness of fit (Fig. 10a, $R^2 \geq 0.07$, $RMSE = 0.12$). There were no significant relationships between $PM_{10}$ and $u_*/u_{*\mathrm{t}}$ (Fig. 10b). This may be because $PM_{10}$ is transported over long distances from upwind regions. Our data showed



that both the total sand transport and the $PM_{10}$ transport were linearly related to the vertical sand transport rate (Fig. 11a, b),
but the sandblasting efficiency ($C_K$, Marticorena and Bergametti, 1995) was much larger (3.82 and 0.25 for total and $PM_{10}$

transport, respectively) than in previous research on shifting sand surfaces ($10^{-5}$ to $10^{-2}$ m$^{-1}$; Marticorena and Bergametti, 1995).
This indicated that sand transport rates were much larger above the gobi surface than above shifting sand, and caused more
sand and dust transport above the gobi surface.

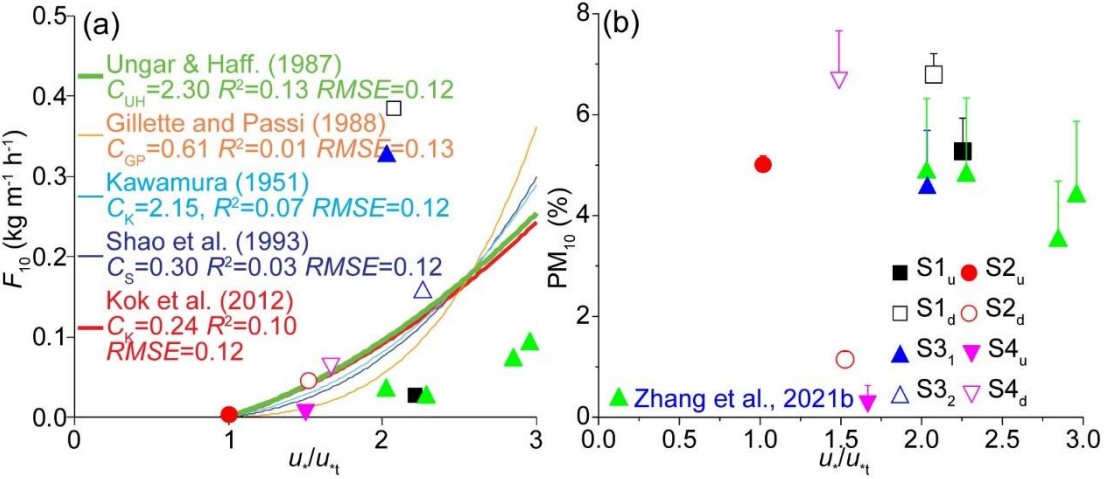

**Figure 10** The relationships between the dimensionless shear velocity ($u_*/u_{*t}$) and (a) sediment transport ($F_{10}$) and (b) the

$PM_{10}$ proportion ($PM_{10}$) measured by the sand samplers. *P<0.05 for all regressions.*

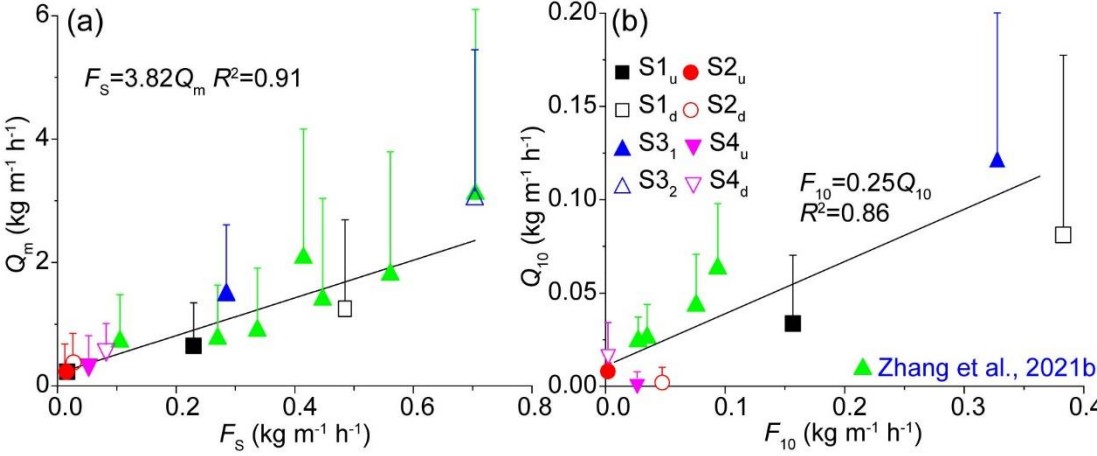

**Figure 11** The relationships between (a) the total vertical sand transport ($F_S$) and the mean sand transport rate ($Q_m$) and (b)
between vertical $PM_{10}$ transport ($F_{10}$) and the mean transport rate for this size class ($Q_{10}$).

### 4.3 Sediment transport above a gobi surface

Grain size strongly affected dust emission and transport (Kok et al., 2012). Kok et al. (2012) showed that aeolian sediment can
be transported by creep (grain size > 500 µm), saltation (63 to 500 µm), and suspension (< 63 µm). Above the gobi surface,





the transported aeolian sediments were mainly fine and medium sands, with a grain size of 63 to 500 µm (74.1±9.3% of the total), followed by silt and clay (grain size < 63 µm, 20.9±9.0%), with the minimum for coarse sand (> 500 µm, 3.9±2.2%) (Fig. 11). Silt and clay were the main dust material during the dust storm, with total contents ranging from 5.5±3.6 to 28.9±8.7%

of the total transported aeolian sediment (Fig. 12a). Surface disturbance increased the availability of silt and clay. At S1, the total silt and clay content was similar for the upwind and downwind sites, at 26.9±8.7 and 26.4±6.7%, respectively, but at S4, the total silt and clay contents were 5.5±3.6 and 28.4±10.1% for the upwind and downwind sites, respectively. This difference showed how the disturbed area of a gobi surface controlled the transported silt and clay content. The wind velocity also affected the transported silt and clay. For sites with a larger wind velocity, such as S3, the strong wind (19.9 m s⁻¹) transported silt and

clay to greater heights, causing a silt and clay content of 12.3±4.6% to a height of 1 m, which is just 45.9% of the value for the undisturbed gobi surface at S1, with the largest wind velocity reaching 17.2 m s⁻¹.

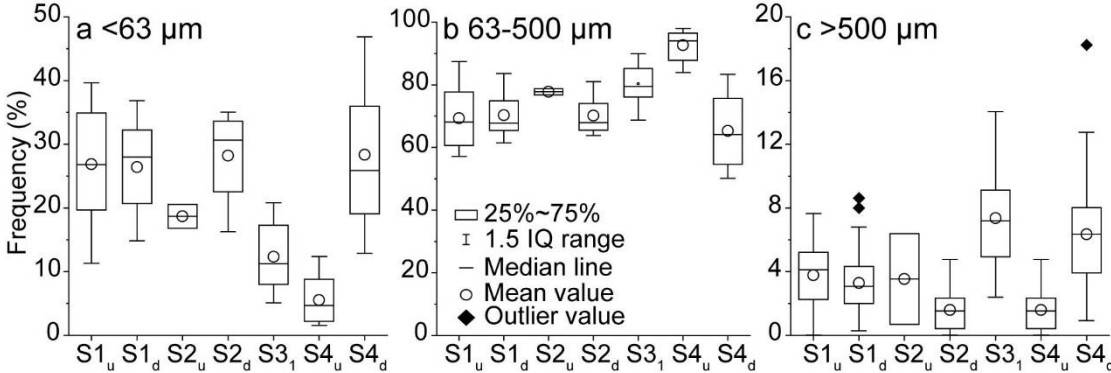

**Figure 12** The frequency of (a) the suspension part of transport (particle diameter < 63 µm), (b) the saltation part (63 to 500 µm), and (c) the creep part (> 500 µm) for the transported sediment at the four field study sites.

Above the gobi surface, we found that the mean frequency of coarse sand ranged from 1.6±1.3 to 7.4±3.0% for all transported aeolian sediment. The frequency of coarse sand was related to the wind velocity, with higher velocities leading to greater creep. For example, at S3, the wind velocity reached 19.9 m s⁻¹, and the mean transported coarse sand frequency was 7.4±3.0%, with the frequency of coarse sand to a height of 1.0 m above the surface reaching 9.6% (Fig. 12c). The saltating coarse sand has more energy than fine sand when it strikes the surface, and this caused more emission of aeolian sediment

from the surface. This can explain the high sand transport rates above the gobi surface.

In our study region, the disturbed gobi surfaces are mainly caused by cars, and the width of the disturbed areas is usually smaller than 5 m (Fig. 1d). As a result of this disturbance, the coarse sand transported above the undisturbed gobi surface can impact particles in the disturbed gobi surface, and this can cause the sediment transport rates above the disturbed gobi surface to become almost 2 times those above the undisturbed gobi surface.





### 4.4 The mechanism of sediment transport above a gobi surface

Sand transport rates are controlled by land surface roughness elements. For example, shifting sand surfaces have almost no roughness elements, whereas gobi surfaces, which are covered by gravel and a soil crust, are much rougher, and this can increase the threshold shear velocity to 0.45 m s⁻¹ above the gobi surface (Zhang et al., 2021a). This is much larger than the threshold shear velocity above a shifting sand surface (about 0.20 m s⁻¹; Kok et al., 2012). Based on field data from the present study and Zhang et al. (2021b), we found that sand transport rates were much larger above the gobi surface than above the shifting sand surface. This can be explained by four main factors: (1) We observed dry lacustrine deposition areas and disturbed gobi surfaces upwind of our study sites (Fig. 1b). This greatly increases sediment transport from upwind, which can be about 139 times the value when there are no upwind dry lacustrine deposits (Zhang et al., under review). (2) We observed a larger wind velocity (a maximum of 19.9 m s⁻¹, Fig. 2) above the gobi surface, which is much larger than the threshold wind velocity above a shifting sand surface (typically about 6 m s⁻¹). Therefore, sand emission and its transport to downwind regions are much greater. (3) The largest sand transport rates were not at the surface (i.e., creep at $z$ = 0 m) (Zhang et al., 2021b), but at 0.05 to 0.07 m above the surface (Fig. 4a). This caused sand to be transported longer distances than above a shifting sand surface because of the longer wind fetch (Zhang et al., 2012). (4) The saltation height of the transported sand ($C$ = 50% of the cumulative sand transport rate) changed with height (Fig. 13a, b) and can reach 0.10 to 0.30 m, versus only 0.03 to 0.04 m for a shifting sand surface (Kok et al., 2012). As a result, sand can be transported longer distances above the gobi surface.

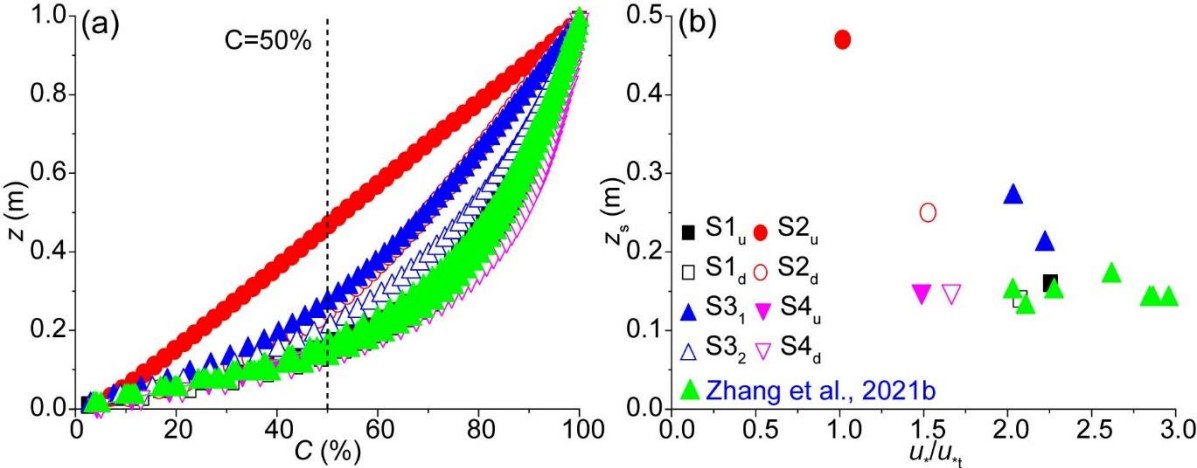

**Figure 13** (a) The cumulative sediment transport rate ($C$) as a function of height ($z$) above a gobi surface. (b) The sand saltation height ($z_s$) calculated using $C$ = 50% of the cumulative sand transport rate as a function of the dimensionless shear velocity ($u_* / u_{*t}$).





## 5. Conclusion

Our results confirmed our research hypothesis: the characteristics of sand transport and the underlying mechanisms for gobi surfaces differed from those for sandy surfaces. This is important because gobis are a major landscape element in northern China. Previous research indicated that gobis are important dust sources in northern China, and we confirmed those results. We filled in gaps in the previous research by obtaining field measurements that confirmed gobis are major dust sources. We obtained the following main results:

1. Wind velocity profiles over the gobi surface during dust storms could be expressed as log-linear functions. The shear velocity ($u_*$) calculated from this function ranged from 0.81 to 1.62 m s$^{-1}$, and the calculated aerodynamic roughness length ($z_0$) ranged from $0.76 \times 10^{-3}$ to $0.81 \times 10^{-3}$ m.

2. Sediment transport rates above a gobi surface can reach 149.4 kg m$^{-1}$ h$^{-1}$, and the transported sediment fluxes can be expressed as a Gaussian peak function, with the maximum transport at a height of 0.08 to 0.17 m. The saltation height above an undisturbed gobi surface can reach 0.10 to 0.30 m. Both the larger sand transport rate and the higher saltation height caused sediment transport over longer distances than above a sandy surface.

3. Both horizontal and vertical sediment transport were related to wind velocity above the gobi surface, but the coefficients were larger than for a shifting sand surface (i.e., transport was greater). Vertical sediment transport was linearly related to horizontal sand transport, and the coefficient was also larger than for a shifting sand surface. The vertical transported PM$_{10}$ content was not significantly related to the horizontal PM$_{10}$ transport rate, but was also not related to wind velocity, suggesting that a significant amount of PM$_{10}$ was transported from upwind regions.

4. The transported sediments above a gobi surface are mainly fine and medium sands, followed by silt and clay. The content of coarse sand is just about 4%, but these transported coarse sand particles impact disturbed gobi surfaces more strongly when they undergo saltation, causing greater sediment emission.

Our findings indicate that aeolian sediment transport rates above gobi surfaces are high, particularly under the strong winds observed in the present study. However, land surface properties, upwind aeolian sources, and higher saltation height above a gravel surface also affected transport. Because our study examined only four sites and only during a single dust storm event, more field measurements will be required to describe sediment transport at more sites and under more normal wind conditions. In addition, it will be necessary to quantify the effects of coverage of a study site by gravel and biological or chemical crusts more accurately to better understand the effects of these factors on sediment transport.

**Data availability**

Data available on request from the authors.



**Author contributions**

ZC wrote, reviewed & edited the manuscript with contributions from all authors. Y contributed to formal analysis. KJ performed the field experiments.

**Competing interests**

The authors declare that they have no conflict of interest.

**Acknowledgments**

We are grateful for the financial assistance obtained from the National Natural Science Foundation of China (41971014, 41930640) , and Gansu Provincial Key Research and Development Program (20YF8WA005). We thank the journal's reviewers for their efforts to improve our manuscript.

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

**Appendix A1:** Layouts of the field experiments.

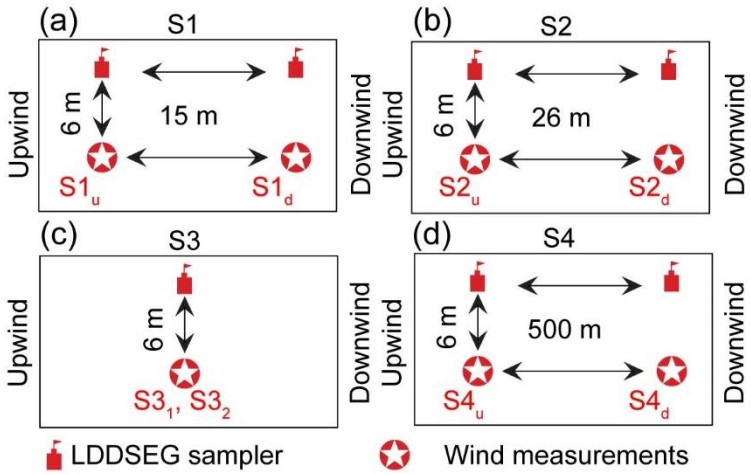


**Appendix A2** Weather conditions during the field experiments.

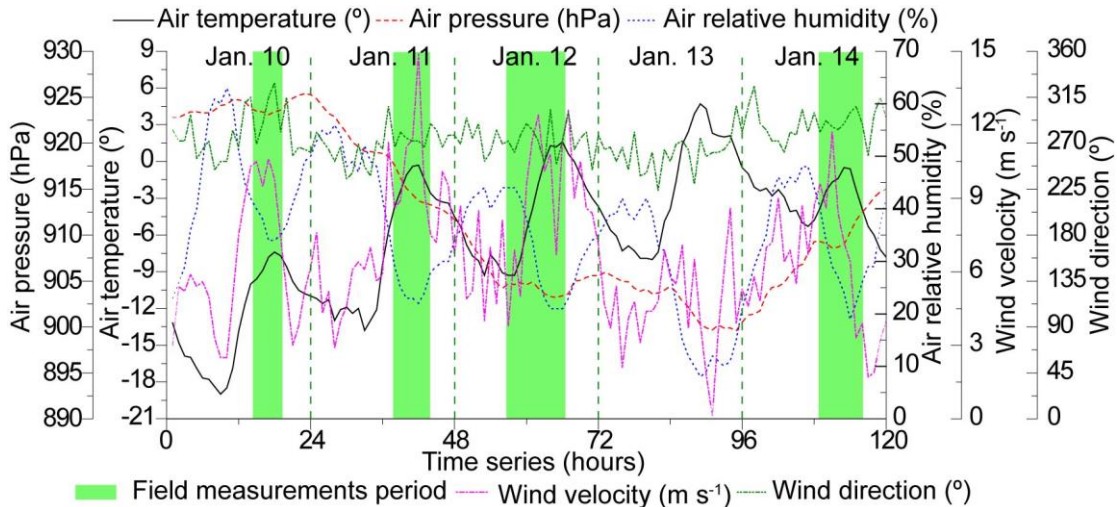





**Appendix A3** Grain-size distributions at sites S2, S3, and S4. Data from site S1 are shown in Figure 7.

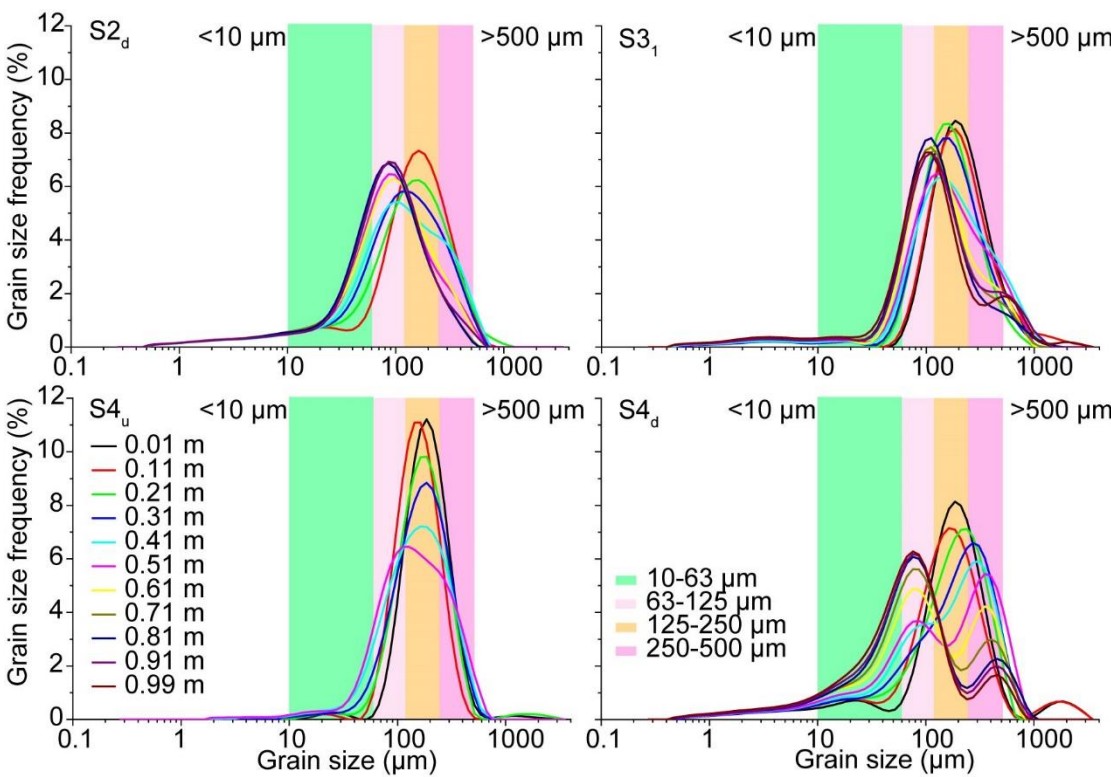

**Appendix A4** Changes in the grain-size distribution with height.

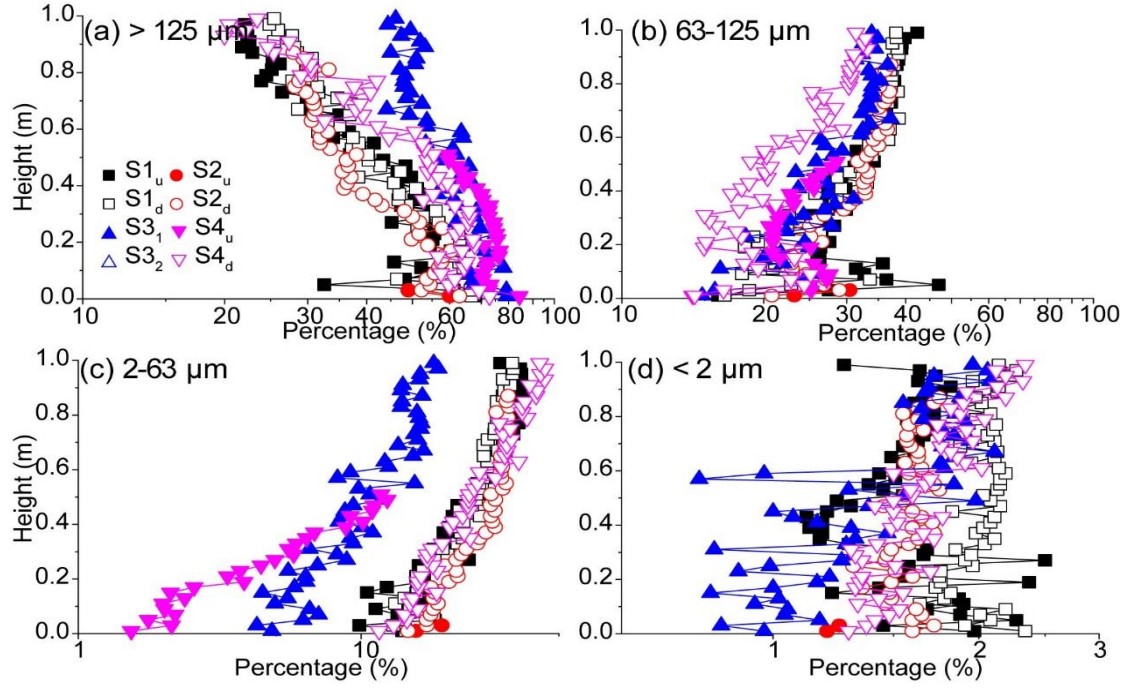