# Peer review of "Characteristics of Aeolian sediments transported above a gobi surface"

_Atmospheric Chemistry and Physics, 2022_

## Author Comment (AC1)

**Dear Mr. Zhang**

This article is a supplement to the sediments transport on different aeolian surfaces. It focuses on dust events in Gobi area, and obtains the coefficient between Q and F by comparing measured data. The results provide support for the current dust research. After reading the manuscript, I look forward to your answers to the following questions: (1) What is the basis for selecting the location of field experiment sites?

Response: The study regions located in the northern China, where had been experienced strong wind erosion and frequency dust storms. Meanwhile, recently government policy had been suggested that new energy will be built above gobi surface. for both of abovementioned reasons, we selected the study region to study Aeolian sediment emission and transport.

(2) Where do the vehicle traffic and animal trampling happen in the study area? How do you quantify these impacts in observations?

Response: During our previous field investigations, we found that some gobi surface had been disturbed by vehicle traffic, which caused by local people graze their animals, and during the railway and road building. Animal trampling is much smaller effect on gobi disturbance, so we did not consider.

For the total disturbed gobi surface, we defined as there are not gravel or crust coverage and all sublayer silt and clay exposed. For the partly disturbed gobi surface, we defined as about 50% land surface gravel or crust had been disturbed.

(3) What may cause the difference in grain-size distribution of transported aeolian sediment?

Response: The difference of grain-size distribution of transported Aeolian sediment should be controlled by 1) up wind available sediment supply, 2) land surface properties, 3) land surface disturbed or undisturbed. If there are rich upwind available sediment supply at the upwind and land surface is undisturbed, the transported Aeolian sediment should be coarser than limited and disturbed gobi surface.

Thank you very much for your comments.

---

## Author Comment (AC2)

Review for Characteristics of Aeolian sediments transported above a gobi surface,
Preprint acp-2022-485

Thank you for coordinating the review of our paper. We have provided responses to review comments in the rest of this letter. We hope that our responses and the resulting changes will be acceptable, but we will be happy to work with you to resolve any remaining issues.

Detailed comments –*Our response is in red text*.
General Comments:

This paper examines aeolian sediment transport on "gobi" desert surfaces in China, which are desert surfaces that are fairly stony, with relatively little exposure of loose sediment (about 50%). The study uses field data from four sites in China's Alxa Plateau to compare aeolian transport rates over gobi surfaces with those over what the authors call "shifting sand" surfaces and to examine how well published parametrizations for horizontal and vertical sediment transport are able to represent the gobi surfaces. The study sites are also used to infer the effects of surface disturbance on sand transport and dust emission.

Overall, the paper adds to the collection of datasets of sediment transport that have been collected at various locations around the world by researchers. There is a paucity of such datasets from Asia and the addition is a welcome contribution in that regard. In terms of impact, the finding that sediment transport rates over a stony surface are much higher than shifting sand surfaces is interesting, but a question arises about whether this is simply a result of very high wind speeds rather than anything that is specific to the stony surface cover. With respect to the quantification of dust (highly suspendible clay and silt, nominally smaller than 15 microns in size), it is unclear if the measurement techniques employed are adequate for accurate assessment of the transport of such particles in suspension. In my view, resolution of this is needed before any significant conclusions can be drawn about suspended dust in this system.

Response:

According to Shao (2008, Physics and Modelling of Wind Erosion), our sampler (similar as Bagnold sand trap) can be used to collect dust particle. He expressed as: *"This type of sand trap can be connected to a vacuum cleaner to make it active. Although such a modification does not ensure an isokinetic flow through the sampling orifice, it allows the sampling of dust particles by means of the sampler."*
Specific Comments:

- Figure 1: All of the panels in Figure 1 are either too small, too difficult to decipher, or both to be helpful. Panel a shows a map with the words "Study Region" but none of the features on the map are labeled and the study region is not clearly marked. Panel b and c are OK, but if panel b was larger, then the underlying topographical

features can be seen better. It's not clear what information is to be extracted from panels d-f. It is hard to see the instrumentation. Suggest replacing these with a plan view of the field measurements and a close up view of the LDDSEG sampler, preferably as deployed in the field.

Response:

We had edited Fig. 1 as following, moved d-f and added LDDSEG samplers in the Appendix A1:

[Figure]

**Figure 1** (a) Location of field experiment sites, (b) the potential sand transport (*DP*, drift potential; *RDD*, resultant drift direction). Field measurement layouts and samplers showed in Appendix A1.

[Figure]

**Appendix A1:** Layouts of the field experiments. (a, d) the S1 field site, (b, e) the S2 field site, and (c, f) the S3 field site, (g) the layout at the at S4 field site, and (h) the LDDSEG vertical

segmented sediment sampler. "u" and "d" represent upwind and downwind sites, respectively.

- Lines 85-95 describe the use of the LDDSEG sampler. It is stated that the sampler captures 86% of the particles being transported. Is this by mass or by number? Is there a difference in this efficiency by size? This is important, because if I understood correctly, the material in these traps is later used to quantify the dust fraction. How well does the sampler capture the dust fraction? Does the sampler have screens? How large are the openings?

Response:

The LDDSEG sampler efficiency is calculated by mass using mixing sand in wind tunnel.
We did not calculate the effect of particle size effect on sampler's efficiency. This sampler can collect all mixed transported particle, so, we think that both dust and sand material can be collected simultaneous. This maybe a question, and we will study it in future.
The collected Aeolian sediment fall down in each collection chamber, and then take it to laboratory for particle size analysis.
There is screen for each sampler's channel.
The detailed information of this sampler showed in Appendix A1.

- Lines 85-95: It's a minor point, but for clarity, were the measurements between S1u and S4d collected simultaneously or was each field site measured at a different time in the Jan 10 -14 period?

Response:

We had added the time in Fig. 1a.

- Lines 95 – 100: There are two concerns I have with this measurement, one minor and one major. The minor concern is that "PM10" refers to suspended particulate matter with aerodynamic diameter of 10 microns or smaller. The Malvern digisizer is not able to estimate aerodynamic diameter. Did you use physical/optical diameter? If so, it would be good to explain that this is not "PM10" in the standard sense of the word. A greater concern that impacts several of the findings later in the paper is that the Malvern digisizer measurement is not ideal for measuring suspended fine dust. If I understand correctly, the instrument is not able to differentiate between dust particles that are free and suspended at the time of collection (i.e., they were in suspension in the air) from those that are simply attached to larger silt and sand particles (i.e., are just along for the ride while a sand particle saltates). If these dust sized particles that are being measured by this technique are not in fact suspended in the air when they are collected, then any conclusions about dust emissions or even horizontal transport of dust will be questionable.

Response:

We had edited PM₁₀ as: grain size smaller than 10 μm in this Lines 98 and related words and figures in the paper. Meanwhile, we deleted the relationships between $PM_{10}$ and $q_{10}$ in Fig. 6b, Fig. 10b.

[Figure]

**Figure 5** (a) The vertical dust emission rates above the gobi surface. (b) The change of the grain size smaller than 10 μm sediment concentration with height on gobi surface.

[Figure]

**Figure 6** The mean grain size smaller than 10 μm sediment concentration collected by the sand samplers. G, gobi desert data from Zhang et al. (2021b); F, fine sand, M, medium sand, C, coarse sand data from Zhang et al. (2017b).

- Lines 123 – 135: It would be good to state explicitly that the sampler had 50 distinct vertical bins.

Response: we had added the following sentence in Line 125.

The LDDSEG sampler simultaneous collected 50 vertical transported sediment samples from 0 to 1.0 m above land surface.

- Line 151: What is the justification for choosing 0.07 m and 0.99 m as z2 and z1? Figure 4 suggests that the lower height (0.07 m ) is located in a part of the profile that is transitional between two different regimes. Equation 9 is meant to be used with concentrations (mass/volume). How does the "horizontal concentration" work within equation 9? Why not convert to a regular concentration?

Response:

According to the Equation (3) and Fig. 4a, we found that 0.07m height is the maximum sediment transport height, and 0.99 m is the minimum sediment transport height for our field measurement. So we used these two heights to calculated vertical dust emission.
We changed as:
We calculated the vertical dust emission ($F$, kg m$^{-1}$ h$^{-1}$) using the method of Gillette et al. (1972) for the collected sand samples:

$$F = ku_* \bar{z} \frac{q_2 - q_1}{z_2 - z_1} \quad (9)$$

where the subscripts 1 and 2 refer to the horizontal aeolian flux in traps 1 and 2 at each of the four sites (0.99 and 0.07 m, refer to the minimum and maximum horizontal Aeolian flux height), respectively. $k$ is the von Karman constant (0.40), $u_*$ is the friction velocity (m s$^{-1}$), and $\bar{z}$ is the mean height (the mean of $z_1$ and $z_2$, which equalled 0.46 m for S1, 0.49 m for S2, and 0.48 m for the others). $F_s$ is the emission rate for all sediment, and $F_{10}$ is the grain size smaller than < 10 μm emission rate.

- Lines 159-162: "…This indicated that wind erosion occurred more easily on disturbed gobi surfaces than undisturbed gobi surfaces." It is difficult to follow the line of reasoning from the previous sentence to this one. In any case the differences in z0 from the two types of surfaces is very small and likely insignificant given the errors in curve fitting.

Response: we edited as:

This indicated that $z_0$ was smaller above disturbed gobi than undisturbed gobi surfaces.

- Lines 190 – 194: What should the reader take away from the difference in the sediment transport rate of change between sites?

Response: we edited as:

The larger $T_h$ indicated that more sediment transported to higher height. We found the ratios were larger above disturbed gobi surface than undisturbed gobi surface, such as at S1, the ratio was 3.0 and 5.9 for $S1_u$ and $S1_d$, respectively. For all disturbed gobi surface, $T_h$ ranged from 4.7 to 6.7.

- Lines 196 – 202: Putting aside the issue mentioned earlier about quantification of dust, it is difficult to say generally that disturbed surfaces are 1.6 to 2.1 times more emissive than undisturbed surfaces without some estimate of what fraction of the surface is disturbed in terms of the area upwind of your sampling instrumentation? Considerable previous work has shown that disturbed surfaces can be orders of magnitude more emissive than undisturbed surfaces, when measurements are highly localized. So, this "ratio" changes with distance from the disturbance area.

Response:

The Aeolian sediment transport above disturbed surface is orders of magnitude more than undisturbed surface, which should relate to the upper wind disturbed land area at the field measurement site.
We added following sentence to explain this difference:
$F_s$ for the disturbed gobi surface ($S1_d$, $S2_d$, $S4_d$) was 1.6 to 2.1 times the value above the undisturbed gobi surface ($S1_u$, $S2_u$, $S4_u$), which indicated that disturbed gobis can provide more vertical sediment flux during dust storms. The lower ratio of $F_s$ between the disturbed gobi surface and undisturbed gobi surface should be related to disturbed gobi area, which controlled sand availability.

- Figure 5b: The x-axis expresses PM10 as a percentage. What does this percentage refer to? It is difficult to relate this directly to a suspended concentration of dust particles as it is currently expressed.

Response: we edited as:

[Figure]

- Line 237: "These results indicate that the sand transport above a gobi surface are much larger than above a shifting sand surface." It does appear that sand transport is higher on the gobi surface, but it is not clear if this is due to the nature of the surface or simply the very high wind speeds.

Response:

We deleted this sentence.

- Section 4.1: I was not able to follow the importance of section 4.1. Suggest adding some explanation of the intent behind the analysis and why the approach was chosen.

Response: we had added following sentences in the paper:

Field-based Aeolian sediment transports were always feasible, therefore, sediment transport rates are estimated by models (Bagnold, 1941; Kawamura, 1951; Pähtz et al., 2011). A convenient way to compare different mass flux relations is to compare the non-dimension mass flux (Kok, et al., 2012). To facilitate the application of our results to modeling aeolian processes, we calculated dimensionless sand transport rate in equation (10). Meanwhile, the initiation of sand transport related to Shields number, we

also analyzed the relationships between dimensionless sand transport rate and Shield number.

- Lines 258 – 270: The discussion of PM10 transport and how it relates to u*/u*T is tenuous because of the previous concern raised about how PM10 was quantified.

Response: we edited this part as:

Both the vertical sand transport rate ($F$) and the transported grain size smaller than 10 μm transport rate ($F_{10}$) increased with increasing $u_*/u_{*t}$ (Fig. 9b, 10). $F$ and $u_*/u_{*t}$ can be expressed using the functions of Kawamura (1951) and Kok et al. (2012) with a moderate goodness of fit (Fig. 9b, $R^2 \geq 0.55$, $RMSE \leq 0.16$; Table 2). $F_{10}$ and $u_*/u_{*t}$ can be expressed using the functions of Kawamura (1951) and Kok et al. (2012), but with a low goodness of fit (Fig. 10, $R^2 \geq 0.07$, $RMSE = 0.12$). Our data showed that both the total horizontal sand transport and the grain size smaller than 10 μm transport were linearly related to the vertical sand transport rate (Fig. 11a, b), but the sandblasting efficiency ($C_K$, Marticorena and Bergametti, 1995) was much larger (3.82 and 0.25 for total and the grain size smaller than 10 μm transport, respectively) than in previous research on shifting sand surfaces ($10^{-5}$ to $10^{-2}$ m$^{-1}$; Marticorena and Bergametti, 1995). This indicated that sand transport rates were much larger above the gobi surface than above shifting sand, and caused more sand and dust transport above the gobi surface.

[Figure]

**Figure 10** The relationships between the dimensionless shear velocity ($u_*/u_{*t}$) and sediment transport ($F_{10}$). $P<0.05$ for all regressions.

- Figure 12: What height do the data in these three panels represent? Also, what is meant by frequency as the y-axis parameter? Is this particle size occurrence frequency?

Response: we edited as:

**Figure 12** The frequency of all 50 transported sediments from 0 to 1 m height (a) the suspension part of transport (particle diameter < 63 μm), (b) the saltation part (63 to

500 µm), and (c) the creep part (> 500 µm) for the transported sediment at the four field study sites.

- Lines 304 – 305: Here again, it would be good to put into context the wind speeds when concluding that transport rates above gobi surfaces are higher than those above shifting sand surfaces. Wind speed is a primary driver of aeolian transport and comparing two locations without comparing the winds they experience does not give a full picture of the inherent transport potential of the surface.

Response:

In the original paper, we had explained the larger sand transport rate above gobi surface should related to larger wind velocity.

- Lines 321-322: The results do indicate a difference in transport magnitudes between gobi surfaces and shifting sand surfaces reported in the literature, but in my view, they do not offer insight into any underlying mechanisms.

Response:

Based on our field experiments, we found that the difference of sediment transport between gobi and sandy surface related to 1) larger friction velocity; 2) larger frequency of transported coarse sand above gobi surface, which these coarse sand impact on land surface and caused more sediment emission from land surface; 3) the larger transported silt and clay further increased the transport rate; 4) higher mean sediment transport saltation height, which caused transported sediment can be transported longer distance . Therefore, above mentioned factor explained the underlying mechanisms.

- Lines 326 – 328: As mentioned above, these friction velocities are quite high and might be the main driver of high transport rates. This is important, but does not necessarily lead to insight into the mechanisms of transport.

Response:

Yes, larger friction velocity related to higher transport rate.
Above gobi surface, the gravel coverage and soil physical crust should be another main reasons for the larger sand transport rate and higher transport height. We added following sentence in the paper:
1. Wind velocity profiles over the gobi surface during dust storms could be expressed as log-linear functions. The shear velocity ($u_*$) calculated from this function ranged from 0.81 to 1.62 m s$^{-1}$, and the calculated aerodynamic roughness length ($z_0$) ranged from $0.76 \times 10^{-3}$ to $0.81 \times 10^{-3}$ m. The larger friction velocity and hard surface properties all caused higher sediment transport rates above gobi surface.

- Lines 335 -337: These talk about the vertical transport of PM10 not being related to wind speed. This would be consistent with the possibility that most of the measured dust particles in the Malvern Digitizer were associated with larger silt and sand particles and not suspended on their own in the air.

Response:

Yes, the grain size measurement may be one of reasons for this relationship, for this reason, we deleted this sentence. We edited this conclusion as:
4 Both horizontal and vertical sediment transport were related to wind velocity above the gobi surface, but the coefficients were larger than for a shifting sand surface (i.e., transport was greater). Vertical sediment transport was linearly related to horizontal sand transport, and the coefficient was also larger than for a shifting sand surface.

Thank you for your efforts to improve our manuscript. We hope that our responses and the resulting changes will be acceptable, but we will be happy to work with you to resolve any remaining issues.

Best regards,
Zhengcai Zhang and all coauthors

---

## Author Comment (AC3)

Review for Characteristics of Aeolian sediments transported above a gobi surface,
Preprint acp-2022-485

Thank you for coordinating the review of our paper. We have provided responses to review comments in the rest of this letter. We hope that our responses and the resulting changes will be acceptable, but we will be happy to work with you to resolve any remaining issues.

Detailed comments

General Comments:

The data collected is interesting, particularly given that such datasets are relatively scarce. However, there are multiple flaws in the application of the methodology and therefore in the results. Also, it is not clear whether the higher sand transport in this study compared to previous ones over shifting sands is just due to a different wind regime (higher wind speeds). In any case, there are not clear insights into the mechanisms that may cause the higher sand transport and the justification seems speculative.

Response:

The higher sand transport related to the both difference of higher wind velocity in the study region and non-erodible land surface above gobi surface (covered by gravel and soil crust).

We had expressed the difference of wind velocity in the study region in Line 314 as:
For the abovementioned factors that control sediment transport above gobi surfaces, we conclude that both horizontal and vertical sediment transport above a gobi surface are controlled by wind velocity (Fig. 8). Because this result is similar to the result for a shifting sand surface, this means that the sediment transport mechanisms above a gobi surface are similar to those above a shifting sand (Kok et al., 2012). The sediment saltation height ($z_s$) was not related to wind velocity (Fig. 12b), which was also true for a shifting sand surface (Kok et al., 2012). However, the different sediment transport rates and saltation heights were caused by coverage of the gobi surface by gravel and a soil crust, which increased the rebound angle for saltating particles above the gobi surface, as Bagnold (1941) suggested.

The uncertainties associated with the measurements are not quantified, and therefore they not taken into account when assessing the results. For example, given that the paper attempts at providing insights into the size distribution, the information provided on the efficiency of the LDDSEG vertical segmented sediment sampler is notably insufficient. Does the 86 % refer to particle mass or number? What is the efficiency per particle size range? Does the efficiency change as a function of wind speed? Typically, passive samplers are much less efficient for small particle sizes. Given that the study also focuses on the dust "PM10" fraction, this aspect is critical, but nothing is discussed in that sense. Another example, is the rather crude derivation of the friction velocity based on only one measurement height.

Response:

This suggestion is similar as Anonymous referee #1. According to the suggestion of the Anonymous referee #1, we explained as following:
The LDDSEG sampler efficiency is calculated by mass using mixing sand in wind tunnel. The efficiency is referring to particle mass.

We did not calculate the effect of particle size effect on sampler's efficiency. This sampler can collect all mixed transported particle, so, we think that both dust and sand material can be collected simultaneous. This maybe a question, and we will study it in future.

The LDDSEG sampler efficiency is similar as most Aeolian sediment sampler, which related to wind velocity, the 86% is the mean value.

As the Anonymous referee #1 suggestions, we changed as grain size smaller than 10 µm transport, but PM10 concentration.

The friction velocity was calculated by wind profiles with five heights, please see Fig. 3.

What is PM10 in this paper? Is it the fraction derived from the Malvern analysis? If the "PM10" fraction is based on the Malvern analysis of the collected samples, how can we know the fraction of dust particles that were already airborne compared to those attached to other particles? I believe this is not possible.

Response:

As the Anonymous referee #1 suggestions, we changed as grain size smaller than 10 µm transport, but $PM_{10}$ concentration. Please see the revised paper.

In eq 9. the gradient method to obtain the vertical dust flux relies on ambient dust concentrations (kg/m3) above the saltation layer. In your study you use c1 and c2 in kg /(m h) which are units of saltation flux at heights within the saltation layer. These units are wrong, and the units of F (vertical dust flux) are also wrong and inconsistent with your already incorrect "concentration" units. Throughout the paper the units of vertical flux are also expressed in kg /(m h) when in reality the vertical dust flux should be in Kg / (m2 h). In other words, you are expressing both concentrations and vertical fluxes with units of horizontal flux. There seems to be a certain lack of understanding of the vertical dust flux, which is defined throughout the paper with very different names (vertical sediment transport, vertical dust flux, vertical sand flux…). I believe that with your measurements you cannot obtain the vertical dust flux appropriately. You are measuring the horizontal fluxes in the saltation layer and at the same time, as mentioned in points 1 and 2 above, the PM10 fraction is likely very uncertain. This implies that the results in many figures (Figures 5, 6, 9 10) are incorrect and that the associated conclusions may be flawed.

Response:

We had edited as vertical sediment transport throughout the paper.

In the conclusions it is stated: "the characteristics of sand transport and the underlying mechanisms for gobi surfaces differed from those for sandy surfaces." However, the paper just concludes that the coefficients that best fit the equations are different than for sand surfaces and does not provide any insight into the mechanisms.

Response:

We had added following sentences in the text:

Specific comments:

Figure 1g mentioned in the caption is missing

Response:

According to the suggestion of the Anonymous referee #1, we had edited Fig. 1 as following:

[Figure]

**Figure 1** (a) Location of field experiment sites, (b) the potential sand transport (*DP*, drift potential; *RDD*, resultant drift direction). Field measurement layouts and samplers showed in Appendix A1.

Line 92: 7 or 8 periods?

Response:

We had edited 'seven' as 'four' for four-day field measurements.

In eq 1 R H and M are not specified. How these factors were calculated? How the final values in Table 4 were calculated?

Response:

We had calculated Eq (1) similar as our previous study (Zhang et al., 2021a), so we did not show the calculation process here. We had written this information in Line 122.

The detailed calculation process showed below.

In gravel deserts, gravels act as roughness elements to produce a sheltering effect on the surface, which effects the value of $u_{*t}$, and is accounted by the function $R$, which, according to Raupach (1992), is calculated following

$$R(\lambda) = \left[\frac{1}{(1-m\sigma\lambda)(1+m\beta\lambda)}\right]^{1/2}, \quad (3)$$

where $\beta = C_R/C_S$, with $C_R$ being the drag coefficient for isolated roughness elements and $C_S$ that for the underlying soil surface, and $\sigma$ is the basal-to-frontal-area ratio of the roughness elements. Following Raupach et al. (1993), we set $\beta = 90$ and $m = 0.5$ in Eq. 3. The roughness (namely, gravel) frontal area index is

defined as

$$\lambda = nbh/s, \tag{4}$$

where $n$ is the number of gravel pebbles within the ground area $s$, and $b$ and $h$ are the typical gravel width and height, respectively. Supposing the gravel pebbles are cylinders, then $n\pi b^2/4s$ is the gravel coverage, and $\sigma$ approx. $b/h$. Our observations show that $h$ approx.5 mm (measured using a Vernier caliper) and $b$ approx. 8 mm (varying between 4 and16 mm), giving $\sigma = 1.6$. Therefore, the frontal area index $\lambda$ is calculated as

$$\lambda = 1.6\,f_g, \tag{5}$$

with $f_g$ being the gravel cover fraction.

We calculated $H$ according to Fécan et al. (1999) as

$$H(w) = \begin{cases} \sqrt{1 + 1.21(w - w_t)^{0.68}} & ,w > w_t \\ 1, & ,\text{otherwise} \end{cases} \tag{6}$$

where $w_r$ is a function of the soil clay content given by $w_r = 0.0014\,f_c^2 + 0.17\,f_c$, with $f_c$ being the percentage of volumetric clay fraction, $w$ is the gravimetric soil moisture for the top-most 0–2-cm soil layer (Fécan et al. 1999; Darmenova et al. 2009; Xi and Sokolik 2015). Gravel deserts in north-west China have little soil moisture. According to their field observations, Liu et al. (2011) found that the gravel desert soil moisture is approximately 0.03 % throughout the year, due to the very small amount of precipitation (< 100 mm per year). Therefore, we set $w = 0.03$ % for all gravel surfaces.

The enlargement function $M$ is another important factor affecting the value of $u_{*t}$ (Marticorena et al. 1997; Batt and Peabody 1999; Ishizuka et al. 2008). In earlier studies, as very limited data concerning the soil crust were available, the value of $M$ was often set to 1 (Lu and Shao 2001). Sharratt and Vaddella (2014) found that the value of $u_{*t}$ for a crusted soil surface increases exponentially with the clay content and soil crust thickness. Following the latter authors, the value of $M$ is computed as

$$M = e^{\,0.026 f_c\, n}, \tag{7}$$

where $n$ is soil crust thickness in mm.

Line 122: Note that you do not infer the friction velocity threshold from observations, you calculate it based on an approximation, which is likely uncertain.

Response:

Yes, we calculated the threshold shear velocity by empirical method, however, as we know, at present, almost all Aeolian research had used empirical method to solve this problem.

Line 123: Why mean and not median diameter?

Response:

The most Aeolian sediment transport study used the mean grain size.

Table 5: 10^-3 instead of 10^3. Note that the differences in roughness are small (probably much smaller than the uncertainty in the method to derive the roughness length)

Response:

We had changed $10^3$ as $10^{-3}$, thanks.

Line 190… I do not understand the description of Figure 4a here. Are you really referring to Figure 4a?

Response:

Yes, it referred to Fig. 4a. Here, for the obvious difference of sand transport above and below threshold height, we used the ratios of the mean sediment transport above and below $T_h$ ($q_u/q_l$, respectively) to express these differences.

Figure 6: you mention PM10 concentration but in reality, it is the fraction of PM10.

Response:

According to the suggestion of the Anonymous referee #1, we did not consider the $PM_{10}$ concentration in the paper, but the grain size smaller than 10 μm, and we had changed all related content in the paper.

Thank you for your efforts to improve our manuscript. We hope that our responses and the resulting changes will be acceptable, but we will be happy to work with you to resolve any remaining issues.

Best regards,
Zhengcai Zhang and all coauthors